# The Effects of Race on Acute Kidney Injury

**DOI:** 10.3390/jcm11195822

**Published:** 2022-09-30

**Authors:** Muzamil Olamide Hassan, Rasheed Abiodun Balogun

**Affiliations:** 1Department of Medicine, Obafemi Awolowo University, Ile-Ife 220005, Nigeria; 2Division of Nephrology, Department of Internal Medicine, Faculty of Health Sciences, University of the Witwatersrand, Johannesburg 2193, South Africa; 3Division of Nephrology, Department of Medicine, University of Virginia, Charlottesville, VA 22908, USA

**Keywords:** race, ethnicity, acute kidney injury

## Abstract

Racial disparities in incidence and outcomes of acute kidney injury (AKI) are pervasive and are driven in part by social inequities and other factors. It is well-documented that Black patients face higher risk of AKI and seemingly have a survival advantage compared to White counterparts. Various explanations have been advanced and suggested to account for this, including differences in susceptibility to kidney injury, severity of illness, and socioeconomic factors. In this review, we try to understand and further explore the link between race and AKI using the incidence, diagnosis, and management of AKI to illustrate how race is directly related to AKI outcomes, with a focus on Black and White individuals with AKI. In particular, we explore the effect of race-adjusted estimated glomerular filtration rate (eGFR) equation on AKI prediction and discuss racial disparities in the management of AKI and how this might contribute to racial differences in AKI-related mortality among Blacks with AKI. We also identify some opportunities for future research and advocacy.

## 1. Introduction

Race can be defined “as a hierarchal human-grouping system, generating social classifications to identify, distinguish and marginalize some groups across nations, regions and the world. Race divides human populations into groups often based on physical appearance, social factors and cultural backgrounds” [1]. Race is a socio-political concept that is inseparably linked to health outcomes in individuals from minority racial status worldwide [2]. Race is not a biological concept.

Racial disparities in clinical outcomes of several health conditions are very common [3]. Such disparities have been reported in AKI, where Black race is associated with a higher risk of AKI compared to non-Black race [4]. Moreover, the prevalence of dialysis-requiring AKI is increasing annually, and Black race is an independent risk factor for initiation of dialysis in AKI patients [4,5,6,7,8]. This phenomenon is thought to be due to the close interplay of clinical, socioeconomic, and genetic risk factors [9], which are driven by structural racial category [10].

Acute kidney injury describes the sudden loss of kidney function, characterized by ≥0.3 mg/dL (≥26.5 μmol/L) rise in serum creatinine within 48 h or an increase in serum creatinine ≥1.5 times over baseline, which is known or presumed within the prior 7 days, or reduced urinary output of < 0.5 mL/kg/h within 6 h [11]. Given that AKI has significant clinical consequences, including higher hospitalization rates, increased hospital length of stay, increased in-hospital and post-hospitalization mortality, and higher risk of progression to chronic kidney disease [12,13,14,15], understanding and addressing how race directly affects patients with AKI is crucial to improving the diagnosis, treatment, and outcomes of AKI in minority racial groups.

In this review, we use the incidence, diagnosis, and management of AKI to illustrate how race is directly related to AKI outcomes, with a focus on Black and White individuals with AKI. We examine racial disparities in COVID-19-related AKI. In particular, we explore the effect of a race-adjusted estimated glomerular filtration rate (eGFR) equation on AKI prediction and discuss racial disparities in the management of AKI and how this might contribute to racial differences in AKI-related mortality among Blacks with AKI.

## 2. Racial Disparities in the Incidence of AKI

Previous studies have reported that Black patients have a disproportionately higher risk of AKI when compared with White patients [4,8,16,17,18,19,20,21,22]. Using self-reported Black and White participants from the Atherosclerosis Risk in Communities (ARIC) study, Grams et al. [4] investigated the impact of race on the incidence of AKI-related hospitalizations. The authors in the large, prospective, community-based studies involving 10,588 African-American and Caucasian participants followed up for 13 years reported significantly higher incidence of AKI in Blacks, with a 30% higher risk in Blacks compared to Whites. After a sequential multivariable model, the risk increased to 50% following the inclusion of age and sex, suggesting that Blacks are at a greater risk of AKI despite their younger age. However, the risks were not significantly different among those participants with major mediators such as obesity, diabetes, or hypertension, which are more common in Blacks compared to Whites. The authors further showed that the relationship between race and AKI was significantly weakened with adjustment for socio-economic indicators such as access to health insurance and total annual family income. 

Similarly, other studies have reported significantly higher incidence of AKI along racial lines among specialized populations, including patients with diabetes [17], trauma [20], and those undergoing procedures such as percutaneous coronary intervention [8,21] and knee surgery [19]. 

In contrast to the findings of Grams et al. [4] in the ARIC study, Muiru and his colleagues, in the Chronic Renal Insufficiency Cohort (CRIC) Study, a multicenter prospective study, observed that Blacks had a 22% greater hazard for AKI hospitalization than the Whites, and showed that the racial disparities in AKI were due to the differences in prehospitalization baseline clinical risk factors such as diabetes, blood pressure, and proteinuria [22]. Thus, this suggested that the increased risk of AKI among Black individuals may be mitigated through targeted screening and tighter control of blood pressure to lower proteinuria. Taken together, all these findings highlight the need for future studies to untie the “Gordian knot” between socio-economic determinants of AKI from those that might be biologic, if any.

Moreover, according to the 2018 USRDS annual data report [23], using data from the Medicare, Clinformatics^TM^, (OptumInsight, Eden Prairie, MN, USA) and Veterans Affairs patient population, a higher proportion of Black patients had AKI compared to Whites or Asians. Among Medicare patients aged 66 years and older, AKI incidence rates were 34.3%, 23%, and 25.9% in Black, White, and Asian patients, respectively, and for Optum Clinformatics™ (OptumInsight, Eden Prairie, MN, USA) patients aged 22 years and older, the incidence of AKI among Black, White, and Asian patients was 9.4%, 7.4%, and 3.3%, respectively. The hospitalized Veterans Affairs populations showed a similar trend—the incidence of AKI was 30.4%, 24.1%, and 24.2% among Black, White, and Asian Americans, respectively. These data therefore highlight differences in AKI incidence by race. Table 1 summarizes the evidence related to the incidence, management, and outcome of AKI in Black and White races.

## 3. Race, APOL1 Status, and Incidence of AKI

Considering the well-established genetic susceptibility for kidney disease among Black populations [33,34], it was not surprising that Privratsky et al. [35] demonstrated a greater-than-two-fold rise in postoperative serum creatinine among Black patients with high-risk APOL1 status undergoing cardiac surgery. However, a previous study by Grams et al. [4] showed that APOL1 high-risk variant was not significantly associated with higher risk of AKI. This study was limited by reliance on billing codes to ascertain AKI, thereby resulting in ill-defined AKI phenotypes. Given that there are several recent reports of association between high-risk APOL1 status and increased risk of AKI among patients infected with severe acute respiratory coronavirus 2 (SARS-CoV-2) [35,36,37,38,39,40,41], further studies are needed to elucidate APOL1 biology, its role in well-defined AKI cases, and its possible modifying effect on progression to CKD after injury.

## 4. Race and COVID-19-Related AKI

Acute kidney injury is a well-recognized complication of SARS-CoV-2 infection, with an incidence rate of 20% among hospitalized patients and greater than 50% among the critically ill patients [42]. Since the first case of SARS-CoV-2 infection which causes coronavirus disease 2019 (COVID-19) was reported in Wuhan, China, racial disparities in the development, severity, and outcomes of this disease have become increasingly reported [43,44,45]. According to the Centers for Disease Control, Blacks had a 1.1 times higher rate of COVID-19 patients, 2.3 times higher rate of COVID-19 hospitalization, and 1.7 times higher COVID-19 mortality rate than Whites [46]. Accordingly, several previous studies have reported that Black patients with COVID-19 had approximately 1.2–2.2 times increased odds of developing in-hospital AKI compared to their White counterparts [47,48,49,50]. Even though the exact mechanism of racial disparities in risk of COVID-associated AKI is yet to be elucidated, some possible mechanisms have been proposed. Genetic polymorphisms involving *ACE2*, *IL-6*, and *AChE* genes are closely associated with higher burden of COVID-19 disease and these gene polymorphisms have been shown to be more common in Black individuals [51,52]. In addition, Black populations are exposed to a higher risk of vitamin D deficiency, and it has been proposed that Vitamin D plays a protective role against COVID-19 infectivity and severity through its ability to modulate the immune system by suppressing T helper-1 functions and potentiating expression of the regulatory T cells, thereby causing less severe cytokine storm [53,54]. Hence, Black patients may face higher risk of cytokine storm and associated systemic and intrarenal inflammation. Table 2 summarizes studies highlighting association between race and COVID-19-induced AKI.

## 5. Race-Adjusted eGFR and AKI Prediction

The use of race-adjusted eGFR is controversial in nephrology [58]. Low eGFR is a strong predictor of AKI and is routinely used by clinicians to predict the risk of AKI following contrast-related procedures [59]. Traditionally, eGFR is estimated using the Modification of Diet in Renal Disease (MDRD) and Chronic Kidney Disease Epidemiology Collaboration (CKD-EPI) formula that includes a correction factor for Black race [60,61]. Remarkably, the inclusion of the race correction factor in the eGFR equation is methodologically flawed because it does not take into account the ancestral diversity among Black patients and patients with mixed racial background, and the fact that it is subjective as a result of self-identification of race. Consequently, in recognition of the fact that race is a social construct with only limited utility in medicine, the National Kidney Foundation and the American Society of Nephrology have now recommended the removal of the race correction factor in eGFR calculation [62], necessitating exclusion of race variables in the new eGFR equations that are currently being developed [63]. In a preprint research study, which used data from the American College of Cardiology (ACC) National Cardiovascular Data Registry (NCDR) CathPCI Registry, Huang and his colleagues demonstrated the impact of removing race from eGFR equations for predicting AKI. The authors showed that the inclusion of the race correction factor significantly underestimated risk for AKI among Black populations (predicted 7.6% vs. observed 10.2%) while marginally overestimating AKI risk among the non-Black population (predicted 7.4% vs. observed 7.1%). However, when the race correction factor was removed, the underestimation of AKI risk was partially corrected among Black patients (predicted 8.2%), and the underestimation was further reduced (predicted 10.1%) by including race as an independent predictor in the AKI model [64]. Hypothetically, eliminating race correction factor from eGFR equations can potentially influence diagnostic and treatment decisions that will take into account each patient’s body habitus, comorbidities, and lifestyle considerations [65,66].

## 6. Racial Disparities in the AKI-Related Mortality

Even though African Americans (Blacks) in the general population are at a higher risk of cardiovascular and non-cardiovascular mortality when compared to Whites [67,68], several reports have suggested that a paradox exists for individuals with AKI, with Black patients having better survival rates than their White counterparts [15,26,69]. Waikar et al. [26], in their study of hospitalized acute renal failure patients between 2000 and 2003 using the Nationwide Inpatient Sample, showed a survival advantage among Black patients with AKI compared to their White counterparts, with Black patients having 16% (95% confidence interval, 10–20%) lower odds for death than White patients with AKI. In our study that included data of 11,567 AKI patients managed at the University of Virginia Health System, we demonstrated that Black patients had lower in-hospital mortality than Whites, with adjusted odds ratio 0.82, 95% confidence interval 0.70–0.96, *p* = 0.0015. Following further analysis, we also extended the findings of the previous workers by demonstrating Black patients had lower 90-day post-hospitalization mortality and better long-term survival advantage when compared to their White counterparts [15]. Similarly, data from the United States Renal Data System (2021 USRDS Annual Data Report) also showed a survival advantage for Black populations compared to White populations after 24 months’ follow-up; Black patients had 38,6% death rates compared to 41.7% death rates in White patients. Previously postulated explanations for this paradoxical finding such as increased access to health care due to medical insurance, “reverse epidemiology” of cardiovascular risk factors, differences in susceptibility to kidney injury, severity of illness, or the response to uremia, warrant additional well-designed investigation [15,26].

Interestingly, like AKI patients, Black patients with end-stage renal disease (ESRD) had also been documented to have lower mortality rates than Whites, and many well-described mediators of paradoxical survival advantage observed among ESRD patients may also be relevant in AKI patients. Most importantly, the similar findings of improved survival for Black patients with AKI and ESRD compared to White patients may suggest that the response to uremia or its treatment may vary by race [26]. Furthermore, previous studies have documented racial differences in biologic processes underlying the pathogenesis of AKI and ESRD, such as interactions between APOL1 genotype and cause of kidney disease [70], abnormalities in endothelial functions [71,72], oxidative stress [73], and inflammation [74]. It is plausible that these biologic factors may account for the paradoxical survival advantage by AKI/ESRD status among Black patients. Further population-based genetic studies in AKI and ESRD patients are warranted to confirm these findings.

## 7. Racial Disparities in the Management of AKI

Patients with AKI are at increased risk of progressive chronic kidney disease (CKD) [75], including ESRD, where they are exposed to additional racial health disparities that impact on AKI outcomes. In addition to facing AKI-related racial disparities, compared to White patients, Black patients receive less intensive hemodialysis and inferior hemodialysis access [76], and are less likely to be offered renal transplantation [77]. Socioeconomic status and limited access to healthcare play a role regarding the racial inequities in the treatment and outcomes of kidney disease. For example, as a result of low income, poor access to healthcare, and lack of health insurance, late referral of Black and/or poor patients with kidney disease for care by a nephrologist prevents these patients from receiving optimal care [78,79]. Figure 1 illustrates how sociocultural factors influences racial/ethnic minority health with implications for progression of AKI to CKD/ESRD.

Notably, attempts have been made recently to improve the quality of healthcare with efforts targeted at eliminating racial and ethnic disparities in health, and this may have major implications for the prevention, diagnosis, and treatment of kidney disease [80,81,82]. Mortality rates in patients with AKI remain high in spite of the skyrocketing cost of the health insurance-funded program. It is conceivable that elimination of these racial disparities would improve outcomes in patients with AKI.

In conclusion, the racial disparities in the incidence, social and biologic determinants, diagnosis, and outcomes of AKI are well-documented and vary between Black and White populations. Disparities in health outcomes as a result of socioeconomic status are discriminating and unjust. Therefore, it is very important to disentangle socioeconomic determinants of AKI such as individual-level income, education, and insurance from those that are biologic risk factors such as diabetes, hypertension, obesity, and cardiovascular disease, which are pervasive in Black populations. Moreover, in order to improve the healthcare of disenfranchised ethnic minorities globally, it is imperative to address lingering racial and ethnic health disparities across the spectrum of kidney disease that continue to perpetuate limited educational opportunities, socioeconomic imbalances, and limited access to healthcare delivery to improve health outcomes of disadvantaged racial minorities with AKI.

## Figures and Tables

**Figure 1 jcm-11-05822-f001:**
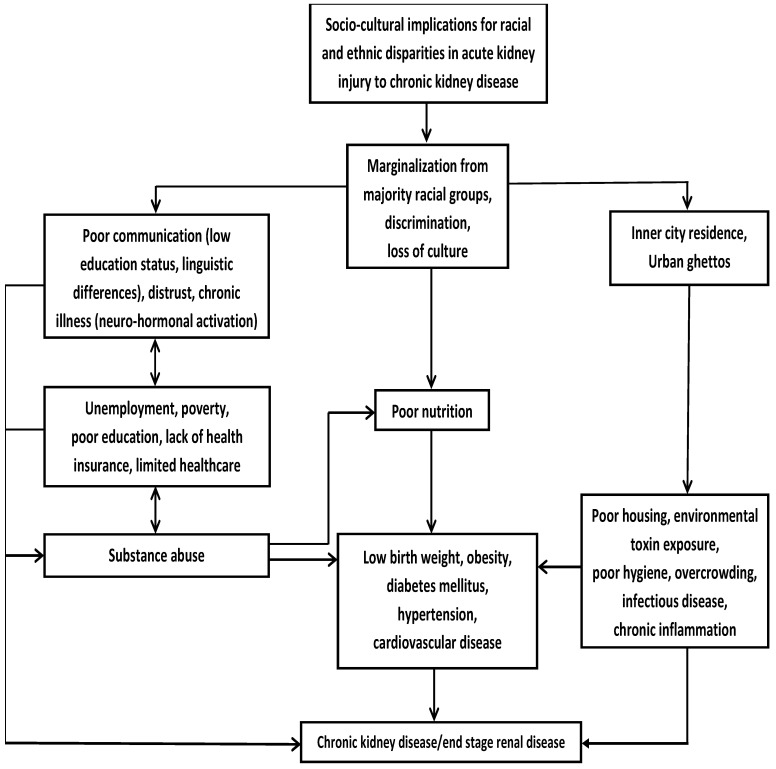
Sociocultural influences on racial/ethnic minority health with implications for progression of acute kidney injury to chronic kidney disease. Adapted with permission from Ref. [6]. 2022, Elsevier.

**Table 1 jcm-11-05822-t001:** Studies summarizing incidence, management, and outcomes of AKI in Black and White races.

Study	Design	Clinical Setting	Population Characteristics	AKI Definition	AKI Rates/Risks (Black vs. White)	Management	Outcomes
Thakar et al., 2003 [24]	Observational cohort study	Database of the Department of Cardiothoracic Anesthesiology, Cleveland	Open-heart surgery patients (*n* = 22,589)	50% reduction or greater decline in GFR relative to baseline or ARF requiring dialysis	2.94% vs. 1.80%	Dialysis (not specified)	ARF and all-cause postoperative mortality
Liangos et al., 2006 [25]	Observational cohort study	National Hospital Discharge Survey database	Hospitalized adult patients (*n* = 29,039,599)	ICD-9-CM codes	ARF more common in Black patients (Data not available)	Dialysis (not specified)	Hospital length of stay, dialysis requirement, hospital death
Xue et al., 2006 [16]	Observational cohort study	Medicare beneficiaries, 1992 to 2001	Hospitalized patients (*n* = 5,403,015)	ICD-9-CM codes	34.4 vs. 22.3 cases per 1000 discharges	Dialysis (not specified)	ARF, sepsis, ICU stay, other acute organ failure, and death within 90 days after admission
Waikar et al., 2007 [26]	Observational cohort study	Nationwide Inpatient Sample	Hospitalized adults (*n* = 15,820,871)	ICD-9-CM codes	3.1% vs. 2.4%	HD and CRRT	In-hospital mortality
USRDS ADR, 2012 [27]	Observational cohort study	Medicare, MarketScan, and Ingenix i3 populations	Hospitalized adult patients (*n* = 1,201,064)	ICD-9-CM codes	44.2 vs. 24.3 per 1000 patient years	PD, continuous HD, intermittent HD, daily HD	Recurrent hospitalization, ESRD, Death
USRDS ADR 2018 [23]	Observational cohort study	Medicare, Optum Clinformatics™, Veterans Affairs population	Hospitalized adult patients (*n* = 854,990)	ICD-9-CM codes and KDIGO	Medicare (34.3% vs. 23.0%), Optum Clinformatics™ (9.4% vs. 7.4%), Veterans Affairs (30.4% vs. 24.1%)	-	Recurrent AKI hospitalization (Optum Clinformatics™), ESRD, and Death
Hsu et al., 2013 [5]	Observational cohort study	Nationwide Inpatient Sample	Hospitalized patients with dialysis-requiring AKI (*n* = 1,095,000)	ICD-9-CM codes	15.6% vs. 10.2%	Dialysis (not specified)	Dialysis-requiring AKI
Grams et al., 2014 [4]	Prospective, community-based cohort study	ARIC study	Individuals aged 45–64 years (*n* = 10,588)	ICD-9-CM codes	7.4 vs. 5.8 cases per 1000 person-years	-	Hospitalization, follow-up time on ACEI/ARB, and Death
Mathioudakis et al., 2016 [17]	Observational cohort study	National Hospital Discharge Survey database	Hospitalized adults with diabetes (*n* = 276,138)	ICD-9-CM codes	6.0% vs. 4.6%	Dialysis (not specified)	AKI, in-hospital mortality, and length of hospital stay
Fisher et al., 2020 [28]	Retrospective observational study	New York City Health System	Hospitalized adult patients with or without COVID-19 (*n* = 9859)	KDIGO	42.0% vs. 9.6%	CRRT, PIRRT, PD, HD	Incident AKI, composite need for RRT or mortality.
Bjornstad et al., 2020 [29]	Secondary analysis study	2012 Kids’ Inpatient Database	Hospitalized pediatric patients aged 1–20 years (*n* = 1,699,841)	ICD-9-CM codes	Risks: 13.4 vs. 12.4	-	AKI
Beers et al., 2020 [30]	Retrospective cohort study	National Inpatient Sample	Patients with pregnancy-related hospitalizations (*n* = 48,316,430)	ICD-9-CM codes	29% vs. 13%	-	In-hospital mortality, adverse discharge, pregnancy-related complications such as miscarriage, preterm labor, and preeclampsia/eclampsia
Hassan et al., 2021 [15]	Observational cohort study	Clinical data repository, UVA Health System	Hospitalized adults with AKI (*n* = 386,342)	KDIGO	3.6 vs. 4.2 cases per 10 person-years	Dialysis (not specified)	In-hospital mortality and post-hospitalization mortality
Heung et al., 2021 [31]	Observational cohort	Perfusion Measures and Outcomes (PERForm) Registry	Adult cardiac surgical patients (*n* = 34,520)	Serum creatinine-based criteria	8% vs. 5%	-	Post-operative AKI
Shah et al., 2021 [32]	Retrospective	Jackson Memorial Hospital database	Patients with ICH hemorrhage and CT angiogram (*n* = 394)	KDIGO	19% vs. 17%	-	AKI
Lunyera et al., 2021 [8]	Observational cohort study	Duke Databank for Cardiovascular Disease	Patients undergoing percutaneous coronary intervention (*n* = 9422)	KDIGO	14% vs. 8%	-	AKI incidence, AKI severity, AKI requiring dialysis, and contrast-induced nephropathy
Muiru et al., 2022 [22]	Multicenter prospective cohort study	Chronic Renal Insufficiency Cohort (CRIC) Study	Participants with CKD hospitalized with AKI (*n* = 2720)	≥50% increase from nadir to peak serum creatinine	6.3 vs. 5.3 per 100 person-years	-	Hospitalized AKI

AKI, acute kidney injury; ARF, acute renal failure; GFR, glomerular filtration rate, CRRT, continuous renal replacement therapy; PIRRT, prolonged intermittent RRT; PD, peritoneal dialysis; HD, hemodialysis; ICD-9-CM, international classification of diseases, ninth revision, clinical modification; ICU, intensive care unit; USRDS ADR, united states renal data system annual data system; ESRD, end-stage renal disease; ACEI, angiotensin-converting enzyme inhibitor; ARB, angiotensin receptor blocker; ARIC, atherosclerosis risk in communities; KDIGO, kidney disease: improving global outcomes; UVA, university of Virginia; ICH, intracranial hemorrhage; CT, computer tomography, CRIC, chronic renal insufficiency cohort.

**Table 2 jcm-11-05822-t002:** Summary of studies showing association between race and COVID-19-induced AKI.

Study	Design	Clinical Setting	Population Characteristics	AKI Definition	Patients with AKI	Risk of AKI	Outcomes
Fisher et al., 2020 [28]	Retrospective observational study	New York City Health System	Hospitalized adult patients with or without COVID-19 (*n* = 9859)	KDIGO	Black (40.5%), White (8.2%), Hispanic (33.5%), Other (17.8%)	Black race was associated with AKI (adjusted odds ratio, 1.7; 95% CI, 1.3 to 2.3)	Incident AKI, composite need for RRT or mortality
Hirsch et al., 2020 [49]	Retrospective observational cohort study	New York Health System	Hospitalized adult COVID-19-positive patients (n = 5449)	KDIGO	Black (20.8%), White (41.0%), Asian (8.1%), Mixed (25.4%), Other (4.2%), Declined (0.4%)	Black race was an independent predictor of AKI (adjusted odds ratio, 1.23; 95% CI, 1.01 to 1.50)	Development of AKI, or RRT and hospital disposition (i.e., discharge or death)
Nimkar et al., 2020 [48]	Retrospective case series	New York City metropolitan teaching hospital	Hospitalized confirmed adult COVID-19 patients (*n* = 370)	KDIGO	Black (43%), White (33.5%), Hispanic (14.5%), Other (8.9%)	African American race showed higher odds of AKI (adjusted odds ratio, 2.1; 95% CI, 1.2 to 3.7	AKI and mortality
Raharja et al., 2021 [55]	Systematic review and meta-analysis	MEDLINE, EMBASE, Cochrane, WHO COVID-19 databases	Seventy-two articles with COVID-19-positive participants (*n* = 17,950,989)	-	White (38%), Black (38%), Asian (5.2%), Hispanic (9.1%), Mixed/other (13%)	Unadjusted analysis showed Black ethnicity had a significantly higher risk of AKI (RR: 1.35, 95% CI: 1.04–1.76), but pooled RR was non-significant (RR: 1.60, 95% CI: 0.89–2.90)	All-cause mortality, hospitalization, critical care admission, invasive mechanical ventilation, extracorporeal membrane oxygenation, and AKI
Bowe et al., 2021 [50]	Observational cohort study	US Veterans Affairs population	Hospitalized COVID-19 patients (*n* = 5216)	KDIGO	Black (53%), White (42%), Other (5%)	Black race was a significant predictor of AKI (adjusted odds ratio, 1.93; 95% CI, 1.69 to 2.20)	All-cause mortality, discharge, need for mechanical ventilation, hospital length of stay
Charoenngam et al., 2021 [47]	Single-center retrospective cohort study	Boston Medical Center	Hospitalized adults with a positive SARS-CoV-2 PCR test (*n* = 1424)	ICD-10-CM codes	Black (39.4%), White (23.1%)	AKI was statistically significantly associated with Black race (adjusted odds ratio, 2.16; 95% CI, 1.57 to 2.97)	In-hospital mortality, intensive care unit admission, hospital morbidities, and inflammatory marker levels
Nugent et al., 2021 [56]	Retrospective cohort study	Yale New Haven Health System network	Hospitalized adult patients with positive COVID-19 and AKI; discharged and did not require dialysis within 3 days of discharge (*n* = 182)	KDIGO	Black (40.1%), White (40.7%), Asian (2.7%), Hispanic (22%), Other (16.5%)	-	Association between COVID-19-associated AKI and eGFR slope after discharge and time to AKI recovery
Bandelac et al., 2022 [57]	Retrospective study	The BronxCare Health System	Hospitalized adults with a positive SARS-CoV-2 PCR test (*n* = 1545)	KDIGO	Black (31.9%), White (1.6%), Hispanic (58.1%), Other (8.4%)	-	AKI, incidence, mortality, stage, and recovery of AKI

AKI, acute kidney injury; KDIGO, kidney disease: improving global outcomes; CRRT, continuous renal replacement therapy; PIRRT, prolonged intermittent RRT; PD, peritoneal dialysis; HD, hemodialysis; ICD-10-CM, international classification of diseases, tenth revision, clinical modification; SARS-CoV-2, severe acute respiratory syndrome coronavirus 2; RR, relative risk.

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
