# Peer review of "The Effects of Race on Acute Kidney Injury"

_jcm, 2022, doi:10.3390/jcm11195822_

Round 1
Reviewer 1 Report
It has been the subject of debate whether the higher incidence of acute kidney injury among blacks is purely due to socio-economic factors or there are biologic determinants.
Many well described mediators of racial disparity in chronic kidney disease may be relevant in acute kidney injury. Interestingly, black patients with acute kidney injury have lower mortality than whites; similarly, the mortality of black patients with end stage kidney disease is lower than that among the white population. The authors may want to address this similarity and speculate on whether there are biologic explanations behind this phenomenon.
The authors need to pay attention to grammatical and spelling errors
Author Response
Thank you for the comments. We were pleased to know that our manuscript was rated as potentially acceptable for publication in JCM.
As suggested, we have addressed the similarities and suggested possible explanation for the racial disparities in AKI and CKD - page 9, line 16-23 to page 10, line 1-4.
"Interestingly, like AKI patients, Black patients with end stage renal disaese (ESRD) had also been documented to have lower mortality rates than Whites, and many well described mediators of paradoxical survival advantage observed among ESRD patients may also be relevant in AKI patients. Most importantly, the similar findings of improved survival for Black patients with AKI and ESRD compared to White patients may suggest that the response to uraemia or its treatment may vary by race (57). Furthermore, previous studies have documented racial differences in biologic processes underlying the pathogenesis of AKI and ESRD, such as interactions between APOL1 genotype and cause of kidney disease (59), abnormalities in endothelial functions (60, 61), oxidative stress (62) and inflammation (63). It is plausible that these biologic factors may account for the paradoxical survival advantage by AKI/ESRD status among black patients. Further population-based genetic studies in AKI and ESRD patients are warranted to confirm these findings."
Regarding the grammatical and spelling errors, we have carefully work through the manuscript and revised all the identified error.
We look forward to hearing from you regarding our submission and to respond to any further questions and comments you may have.
Kind Regards.
Reviewer 2 Report
The authors describe the effect of race on AKI development focusing on the black population. The review is nicely written. Since the title does not particularly focus on the black race, I suggest adding a paragraph describing the risk of AKI in Asian population and comparing it with the risk among black (African Americans) and Caucasians.
Author Response
Thank you for your comments. We were pleased to know that our manuscript was rated as potentially acceptable for publication in JCM.
As suggested, we have included a paragraph to compare the risk of AKI among Blacks vs Whites vs Asians - Page 5, paragraph 3, line 13-22.
"Moreover, according to the 2018 USRDS annual data report (23), using data from the Medicare, ClinformaticsTM and Veterans Affais patients population, a higher proportion of Black patients had AKI compared to Whites or Asians. Among Medicare patients aged 66 years and older, AKI incidence rates were 34.3%, 23% and 25.9% in Black, White and Asian patients, respectively, and for Optum Clinformatics™ patients aged 22 years and older, the incidence of AKI among Black, White and Asian patients were 9.4%, 7.4% and 3.3%, respectively. The hospitalized Veterans Affairs populations showed a similar trend – the incidence of AKI were 30.4%, 24.1% and 24.2% among Black, White and Asian Americans, respectively. This data therefore highlights differences in AKI incidence by race. Table 1 summarizes the evidence related to the incidence, management and outcome of AKI in Black and White races."
We look forward to hearing from you regarding our submission and to respond to any further questions and comments you may have.
Thank you.
Reviewer 3 Report
This review article on the effects of race on AKI summarized the recent advance in the relationship between racial disparities and clinical outcomes of AKI. The manuscript is well and concisely written.
As a review article, the review should be more extensive. Many studies have shown that race have effect on AKI outcomes. Can you expand the number of references and add a table summarizing the incidence, management and outcome of AKI in black and white races ?
AKI frequently complicates the course of COVID-19. There are several papers reported that race is a risk factor of COVID-19 induced AKI. Please also add a table listing these studies and summarize their findings.
Line 48 and 65: "acute kidney injury" should use abbreviation.
Line 100-102: as the author mention that "Given that there are several recent reports", relevant references should be added.
Line 102 and 106: SARS-COV-2 -> SARS-CoV-2
Line 164 with adjusted odds ratio: 0.82; 95% confidence interval 0.70–0.96, p = 0.0015) > need another bracket
Author Response
Thank you for all the comments. We were pleased to know that our manuscript was rated as potentially acceptable for publication in JCM.
Comment:
"Can you expand the number of references and add a table summarizing the incidence, management and outcome of AKI in black and white races ?"
Action:
We have expanded the list of references and added Table 1 as suggested - pages 23 - 26
Comment:
"There are several papers reported that race is a risk factor of COVID-19 induced AKI. Please also add a table listing these studies and summarize their findings."
Action:
We have included table 2 as suggested and updated the list of references - pages 27-28
Comment:
"Line 48 and 65: "acute kidney injury" should use abbreviation."
Action:
Revised accordingly
Comment:
"Line 100-102: as the author mention that "Given that there are several recent reports", relevant references should be added."
Action:
We have included reference nos 26-32.
Comment:
"Line 102 and 106: SARS-COV-2 -> SARS-CoV-2"
Action:
Revised as suggested
Comment:
"Line 164 with adjusted odds ratio: 0.82; 95% confidence interval 0.70–0.96, p = 0.0015) > need another bracket"
Action:
The sentence has been revised.
We look forward to hearing from you regarding our submission and to respond to any further questions and comments you may have.
Thank you.
Round 2
Reviewer 3 Report
The authors have satisfactorily responded to all my questions and made the necessary changes to the manuscript.